# Influence of Sensor Position and Low-Frequency Modal Shape on the Sensitivity of Vibro-Acoustic Modulation for Impact Damage Detection in Composite Materials

**Gabriela Loi** [1], **Francesco Aymerich** [1] and **Maria Cristina Porcu** [2,*]

1   Department of Mechanical, Chemical and Materials Engineering, University of Cagliari, 09123 Cagliari, Italy; gabriela.loi@unica.it (G.L.); francesco.aymerich@dimcm.unica.it (F.A.)
2   Department of Civil and Environmental Engineering and Architecture, University of Cagliari, 09123 Cagliari, Italy
*   Correspondence: mcporcu@unica.it

**Abstract:** Very sensitive structural health monitoring methods are needed to detect barely visible impact damage in composite materials. Based on extracting non-linear modulated components from the frequency response of the damaged system, vibro-acoustic modulation (VAM) has shown to be effective in identifying the presence of damage at its early stage. A decisive role in the success of this technique is played by the choice of the high-frequency probe and the low-frequency pump sinusoidal signals that simultaneously excites the system. This study explores how the position of the sensing transducer, with respect to the modal shape of the pump excitation, may influence the sensitivity of the VAM technique for impact damage detection in composite laminates. This aspect has been scarcely investigated in previous research works, as other studies have focused more on the role of the probe frequency. Here, VAM tests were performed on a composite beam by using a frequency-swept pump vibration simultaneously with a high frequency probe excitation. The results of the experimental tests indicate that the VAM technique is capable of clearly revealing the presence of impact damage only when the sensor is placed on appropriate locations, which are directly related to the shape of the deformation activated by the applied excitation. These results suggest the adoption of low frequency excitations that activate multiple modal shapes to improve the effectiveness and reliability of VAM approaches.

**Keywords:** structural health monitoring; impact damage detection; vibro-acoustic modulation; swept pump excitation





## 1. Introduction

Ageing deterioration processes or sudden injuring actions are some of the causes that may affect the integrity of structural materials during their service-life, thus it is important to have the reliable methods of structural health monitoring (SHM). To detect the presence of damage, SHM may focus on the global performance of the considered system. The vibration-based techniques do this by exploiting the changes that some modal properties (frequencies, mode shapes, damping, curvature of the mode shapes, or of the frequency-response-function) undergo in the damaged structure. Although widely used to monitor the health of mechanical and civil structures made of different materials [1–6], such methods are generally able to detect macro-level damage [5], while showing a low sensitivity to smaller local defects [1].

SHM methods that rely on extracting non-linear features from the system vibrational response have been found to be more sensitive to localized small defects like cracks in steel [7], diffused microcracks in concrete [8], or barely visible impact damage in composite laminates [9–11].

Among the non-linear techniques, the vibro-acoustic modulation (VAM) has gained increased attention in the last few decades because of its sensitivity to different types of damage in various classes of materials [7,12–19]. The VAM technique is based on the simultaneous introduction in a component so that it can be monitored at a low-frequency wave (pump excitation) and a high-frequency wave (probe excitation). The pump vibration excites the damage, which in turn perturbs the sensing probe wave, leading to the appearance of modulation sidebands around the probe frequency in the spectrum of the system response. Therefore, the onset of sidebands is assumed as an indicator of the presence of damage, or other kinds of sources of nonlinearity, in the material.

In view of its sensitivity to small defects, VAM has been exploited to identify damage induced in composite materials by low-velocity impacts, which typically consists of localized delaminations and matrix cracks that may be difficult, if not impossible, to detect by visual inspection. The vulnerability to low-velocity impact loadings is in fact a limitation to the otherwise excellent performance of this class of materials for a wide range of engineering applications. The potential of the VAM approach for detection of impact damage in composite structures was first assessed in 2005 by Meo and Zumpano [20]. The examined structure was a sandwich panel with a laminated composite skin, and the damage was introduced by impacting the plate with a sharp object, resulting in a visible hole, 1 mm in diameter and 5 mm deep. Successively, Aymerich and Staszewski [21,22] explored the sensitivity of VAM to different severities of impact damage (ranging from barely visible damage to the onset of penetration) in a thin carbon-epoxy laminate. The experimental analyses showed that the amplitude of the sidebands can be related to the amount of damage and that the frequency of the pump excitation plays a key role in the effectiveness of the technique. Similar trends were observed by Klepka et al. [23], who applied the VAM to identify the presence of impact-induced damage in a chiral-core composite sandwich, finding that the choice of the pump frequency greatly affects the performance of the technique. The effect of both the pump and the probe frequency on the vibro-acoustic response of an impacted multidirectional laminate was investigated in [24] by a combined experimental and numerical analysis. It was found that changes in the pump or the probe frequency may result in a significant variation of the nonlinear content of the system response; in particular, higher sideband amplitudes were observed for pump frequencies that excite the delaminated interfaces with predominant out-of-plane motion and for probe frequencies that match the local resonance of the delaminated region.

The dependence of the VAM performance on the frequencies of the pump and probe excitation has been reported in the literature not only for impact damage in composite laminates, but also for other classes of materials, such as metals and concrete, and for a variety of sources of nonlinearity, including fatigue cracks in metal components [13,25–27]; bolt looseness in mechanical joints [28]; artificial scatterers or kissing bonds [29,30]; and diffuse micro- and macro-cracking [17].

However, most of the previous research work on this dependence has mainly addressed the role of the probe frequency on the performance and reliability of the VAM technique. A clear correlation between the frequency of the probe excitation and the sensitivity of the technique was observed by Duffour et al. [13] on cracked steel samples subjected to probe excitations with frequencies ranging between 50 and 230 kHz. Cracked steel beams were also examined in a series of VAM tests by Yoder and Adams [25], who reported that the magnitude of the sidebands increased with the increase of the amplitude of the spectral response at the sideband frequency. Their experiments also showed that the intensity of the modulation was strongly dependent on the deformation shapes of the system associated with the probe and pump frequencies, thus indicating that some preliminary knowledge of the deformation response of the system is needed for an appropriate placement of sensors and actuators under fixed frequency harmonic excitations. To overcome this need, the authors proposed to use a swept probe excitation, so as to inspect a wide range of probe frequencies in a single test. A sweep signal was also used for probe excitation in [26,27] to examine the influence of the probe frequency on the capability of the

technique to detect fatigue cracks in aluminum beams. The experimental results indicated that the modulation reached a peak when one of the sideband frequencies coincided with a resonance of the system, thereby suggesting that the best crack sensitivity is achieved when selecting a probe frequency that is shifted from the resonance frequency by a distance equal to the applied pump frequency. The amount of modulation was also affected by the location of the crack, with cracks located on nodal points of the deformation shape generated by the probe excitation resulting in very low sideband amplitudes.

Much less research work has been done to investigate the role of the frequency and vibration shape of the pump excitation on the reliability of the VAM technique. Donskoy et al. [31] examined the sideband response of fatigued steel samples subjected to a pump excitation applied by a hammer impact and consisting of the superposition of multiple bending modes. They observed that the nonlinear sideband responses were greatly dependent on the mode of vibration and correlated this behavior with the different levels of perturbation introduced in the cracked region by the different vibration modes. A parametric analysis of the effect of the pump frequency on the VAM performance was performed in [29] on unidirectional carbon/epoxy laminates containing artificially intro- duced defects (delaminations and liquid layer bonds). The study shows that the results of the VAM vary significantly with the pump frequency and that, depending on the type of the defect, damage may remain undetected for certain pump excitation frequencies. A clear influence of the pump frequency on the magnitude of the spectral sidebands was also observed in aluminum specimens with a fatigue crack for pump frequencies in the ultrasonic range (50–150 kHz) [15]. The dependence of the intensity of modulation on the pump frequency was also reported for fatigue cracked aluminum rods in [32]. This effect was attributed to the different amounts of relative motions occurring at the crack interfaces for the different pump frequencies. In this respect, it may be worth noting that most studies highlight the importance of exciting the damaged area with sufficient strain energy to activate its nonlinear behavior [13,26,27,29–31,33]. This necessity has suggested the adoption of VAM approaches based on the application of multiple pump excitation modes to reduce the chance of having a node of the deformation shape positioned at the damage location [15,25,30,31,34].

As a general remark, it is also important to observe that the features and the mor- phology of cracks in metals or artificially inserted defects in composites are drastically different from those of the damage generated by real impacts in composites; the indications provided by the studies summarized above may thus not necessarily apply to other forms of damage and should be verified for the nonlinearities introduced by the actual impact damage mechanisms occurring in composite laminates.

Motivated by this background, this study aims to explore a specific aspect of the practical application of the VAM technique that has not been examined in detail in previous studies and that concerns the location of the sensor with respect to the characteristics of the low-frequency excitation. In particular, this paper examines how the spatial and frequency features of the pump vibration affect the performance of the VAM technique for detection of barely visible impact damage in composite laminates. Special attention has been paid to characterize the correlation existing among the deformation shape under the pump excitation, the position of the sensing transducer and the sensitivity of the technique. The VAM tests were performed on a laminated composite beam excited by a frequency-swept pump vibration and a single harmonic probe wave. The tests were first conducted on the beam in the pristine (undamaged) condition and then, after the introduction of damage, by a low-velocity impact. The location of the sensor used for acquiring the response of the beam was chosen by a preliminary FE analysis, in order to ensure the placement of the sensor in nodal or antinodal regions of the modal deflections activated by the swept pump excitation. Time-frequency analyses were carried out to evaluate the performance of the technique and to assess possible approaches to improve the reliability and the robustness of the technique for detecting damage in composite materials.

## 2. Vibro-Acoustic Modulation Technique

In its original conception, the nonlinear vibro-acoustic modulation (VAM) technique [35] is based on concurrently exciting the system through two pure-tone harmonic signals: a low-frequency (LF) wave which is supposed to perturb the damage (pump signal) and a high-frequency (HF) wave, used to sense the damage (probe signal). The frequency of the HF signal is typically chosen in the ultrasonic range while the LF signal is usually tuned to one of the lower natural frequencies of the inspected system, cf. e.g., [23,24]. The dynamic modulation of the two impinging signals is expected to make the non-linearities due to damage arise in the response of the system. This is revealed by the occurrence of a pattern of extra components (sidebands) rising around the main high frequency in the power frequency spectrum of the damaged system. Therefore, the onset of sidebands is assumed as an indicator of the presence of damage while the sideband amplitude is supposed to be related to the defect severity.

The non-linear acoustical techniques are much more sensitive to barely visible defects, even when the latter are much smaller than the wavelength of the interrogating signal [7]. In principle, a nonlinear response is assumed for damaged materials, while the undamaged material is supposed to have a linear elastic behavior.

Let $v_{LF}(t)$ and $v_{HF}(t)$ be the steady-state responses of a linear system to each of the two *LF* and *HF* input signals:

$$v_{LF}(t) = V_{LF} \, \sin(2\pi f_{LF} t + \varphi_{LF}) \tag{1}$$

$$v_{HF}(t) = V_{HF} \, \sin(2\pi f_{HF} t) \tag{2}$$

Here, $V_{LF}$ and $f_{LF}$ are the LF pump amplitude and frequency, $V_{HF}$ and $f_{HF}$ are the amplitude and frequency of the HF probe signal, while $\varphi_{LF}$ denotes a phase angle. When a linear (undamaged) material is concerned, the superposition principle holds and thus the steady-state response of the system to the two input signals (1) and (2), concurrently applied, would be:

$$v(t) = v_{LF}(t) + v_{HF}(t) = V_{LF} \, \sin(2\pi f_{LF} t + \varphi_{LF}) + V_{HF} \, \sin(2\pi f_{HF} t) \tag{3}$$

In this case, the system response is characterized by a power frequency spectrum where only the two exciting frequencies $f_{LF}$ and $f_{HF}$ appear (Figure 1).

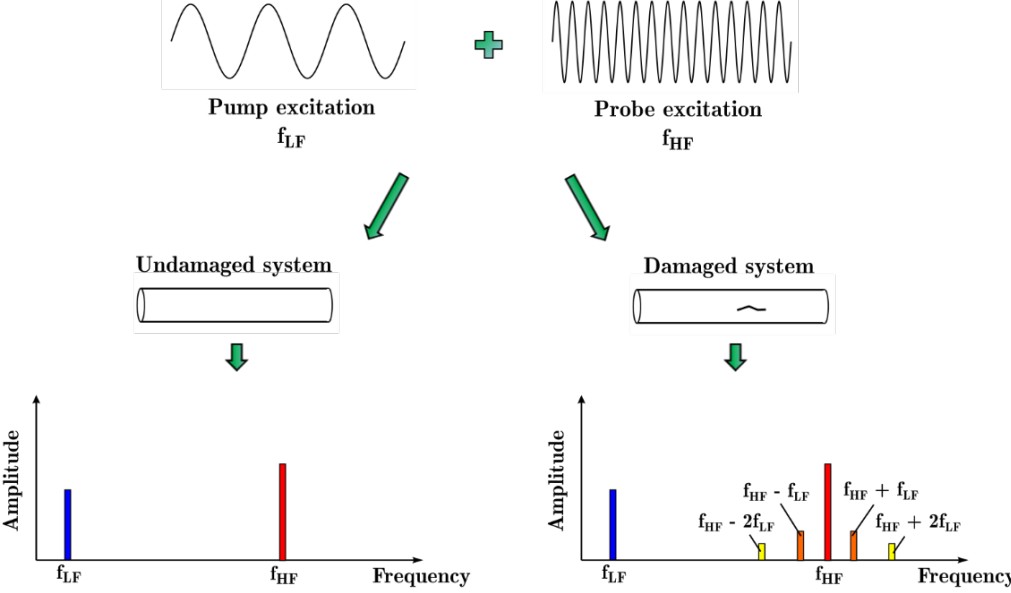

**Figure 1.** Schematic representation of the nonlinear vibro-acoustic modulation (VAM) technique.

On the contrary, when the material is damaged, nonlinear mechanisms (such as contact, decohesion and friction) occur in the damaged area that cause a nonlinear response against inputs [7,12]. This results in the generation of new waves of other frequencies [36], such as higher- or sub-harmonics and sidebands. In particular, modulation sidebands (Figure 1) typically occur in pairs (left and right sidebands) at frequency values corresponding to:

$$f_{SB_n} = f_{HF} \pm n\, f_{LF} \tag{4}$$

where $n$ is a positive integer number.

It is worth noting that, although in theory the generation of new waves and frequencies is an indicator of the presence of damage, sidebands can be observed even in the spectrum of the intact system, due to other kinds of nonlinearity (for instance, inherent nonlinearities of the material, boundary conditions, equipment) [24,29].

Many sources of nonlinearity may in fact affect the response of a material, ranging from intrinsic imperfections of the material (microdefects and imperfections at the micro scale) to non-symmetric thermo-elastic and dissipative mechanisms at the damaged interfaces (contacts, rubbing, friction, adhesion hysteresis, bilinear stiffness, clapping) [12,14,36]. Nonlinear effects may involve different nonlinear mechanisms related to damage and vice versa. Several theoretical models have been proposed in the literature to study such mechanisms, although none of them have gained wide acceptance [14,36]. For instance, an overview of models with bilinear stiffness accounting for the crack opening/closing mechanism is presented in [36], while a more comprehensive three-dimensional model is proposed in [37], where defects were modelled through interfaces and the nonlinearity due to multiple mechanisms like contact, friction and adhesion was accounted for in the stress/strain fields.

## 3. Composite Sample and Experimental Setup

The sample used in the current study, shown in Figure 2, was a 520 mm × 59 mm × 2.2 mm laminated composite beam with $[0/90]_{3s}$ stacking sequence. The beam was cut from a laminate manufactured from Seal Texipreg® HS160/REM carbon/epoxy unidirectional prepreg plies (0.17 mm nominal thickness) and cured in an autoclave at a pressure of 8 bar and a maximum temperature of 160 °C. Before testing, the sample was ultrasonically scanned to exclude the presence of manufacturing defects.

The beam was instrumented with a low-profile piezoceramic transducer (PI Ceramic PIC 151 with a diameter of 10 mm and a thickness of 1 mm) glued to the sample with a two-component epoxy adhesive. The beam was connected to a Bruel and Kjaer 4809 electrodynamic shaker through a threaded aluminum stud bonded to its bottom face.

Both the modal and VAM tests were conducted using the experimental setup shown in Figure 2. The shaker used to vibrate the sample (A1 in Figure 2) was driven by a signal provided by a TTI-TGA1241 function generator and amplified by a Bruel and Kjaer 2706 power amplifier. The piezoceramic transducer bonded at location A2 was used as a high frequency actuator. The actuator was driven by a signal supplied by a TTi-TG5011 function generator and amplified with a FLC F20A amplifier. The system response was recorded at sensor locations S1 and S2 using a Brüel and Kjær 4375 charge accelerometer, whose output was fed into a Bruel and Kjaer 2635 charge amplifier and acquired with a 14 bit, 100 MSa/s, PC-controlled oscilloscope (Cleverscope CS328A). The positions S1 and S2 were chosen through a preliminary modal analysis as described in Section 4.

A drop-weight testing machine equipped with a 2.34 kg impactor provided with a 12.5-mm hemispherical indenter was used to introduce impact damage in the sample. The composite beam was impacted at the location indicated in Figure 2 with an energy of 1.9 J, which induced a typical barely visible impact damage (BVID). A non-destructive penetrant-enhanced X-ray inspection was carried out after the impact to characterize the internal damage, which mainly consisted of a combination of matrix cracks and delaminations (Figure 3), with a projected damage area of about 40 mm². A minor fiber fracture could also be observed on the impacted face at the boundary of the indentation area. Fiber failures

may be distinguished from matrix cracks by their thick and jagged paths, which clearly differ from the extremely thin and straight lines of 0° and 90° matrix cracks.

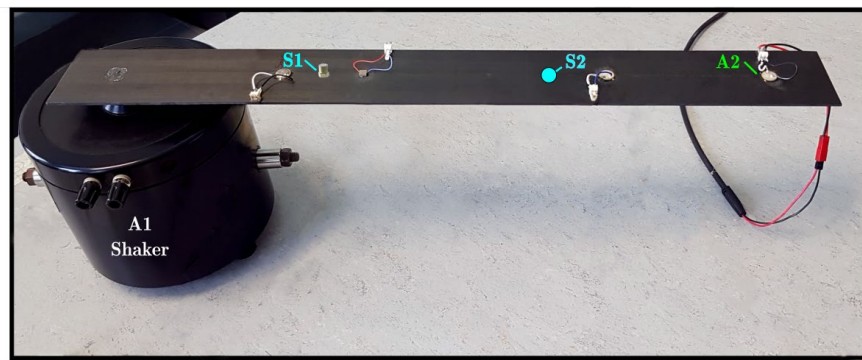

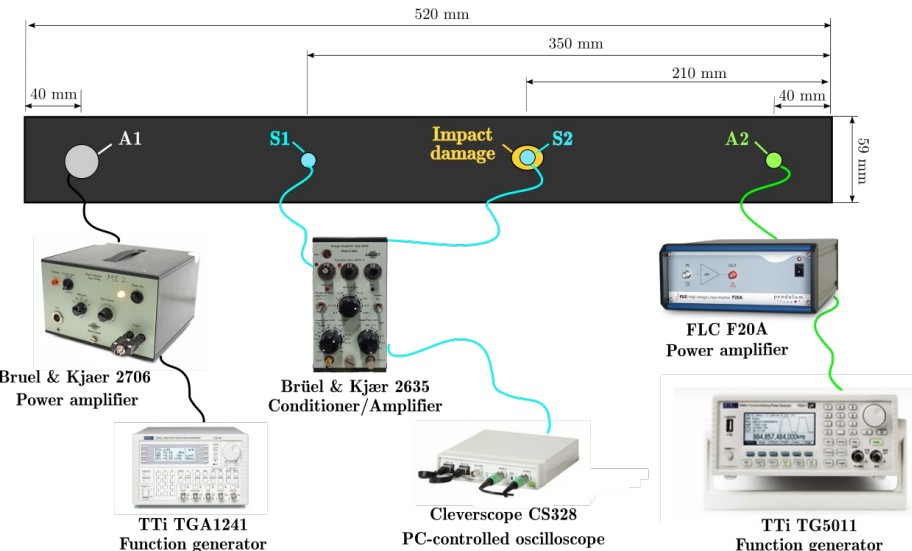

**Figure 2.** Experimental setup.

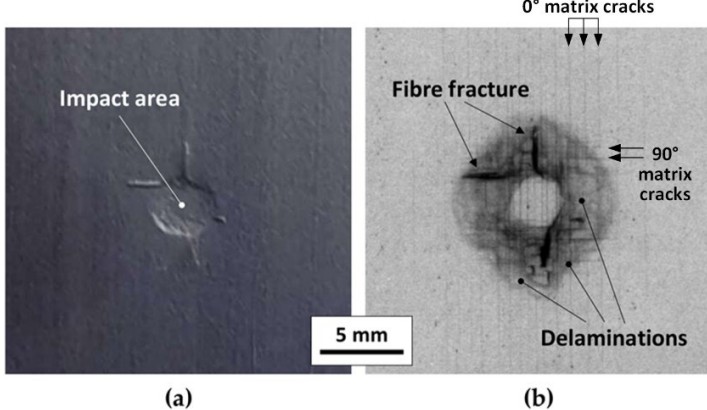

**Figure 3.** Surface indentation (**a**) and radiographic image of internal damage (**b**) induced by the low-velocity impact.

Because of the small size of the delaminated area and the presence of only minor fiber failure, resonance frequency changes of less than 1.5% were measured after the introduction of damage. Changes of this amount may fall within the error ranges of this kind of measurements, making it arduous to detect the presence of damage with sufficient confidence by simple monitoring of resonance frequencies.

## 4. Modal Analysis

A 3D finite element model of the composite sample was preliminary developed in Abaqus to characterize the dynamic response of the system and identify the resonance frequencies and the associated mode shapes of the examined beam. Here, 8-node brick elements (C3D8R) and 10-node tetrahedral elements (C3D10) were used to model the beam and the bonded aluminum stud, respectively. To properly reproduce the stacking sequence, one solid element was used for each ply across the thickness. The elastic properties used in the FE simulations are reported in Table 1. The main properties of the unidirectional composite layer used in the model were obtained by tests on $[0]_{10}$, $[90]_{10}$ and $[+45/-45]_{2s}$ laminates.

**Table 1.** Mechanical properties of the unidirectional composite layer and of aluminum used in the FE simulations.

| Composite Properties | Symbol | |
|---|---|---|
| Mass density | $\rho_{CFRP}$ | 1500 kg/m$^3$ |
| Young's moduli | $E_x$ | 95 GPa |
| | $E_y = E_z$ | 7.5 GPa |
| Shear moduli | $G_{xy} = G_{xz} = G_{yz}$ | 4.0 GPa |
| Poisson's coefficients | $v_{xy} = v_{xz} = v_{yz}$ | 0.26 |
| **Aluminum properties** | **Symbol** | |
| Mass density | $\rho_{Al}$ | 2760 kg/m$^3$ |
| Young's modulus | $E$ | 70 GPa |

Among the modal shapes identified by the FE model, the flexural modes corresponding to the natural frequencies of 167 Hz, 313 Hz, and 504 Hz, as shown in Figure 4, were chosen to guide the choice of sensor location for the subsequent VAM tests. The choice of these mode shapes allowed the identification of a position (S1 in Figures 2 and 4) that lies close to antinode regions for the 167 Hz and 313 Hz modal frequencies but coincides with a node for the 504 Hz frequency, thus enabling us to examine how the location of the sensor with respect to the spatial characteristics of the pump vibration affects the sensitivity of the technique.

The natural frequencies obtained by the FE simulations were verified by an experimental modal analysis performed on the system. To this purpose, the composite beam was excited through a linear sine sweep ranging between 1 Hz and 1000 Hz, while the response of the system was measured by means of the accelerometer positioned at the location S1 previously identified through the FE analysis. In total, 10 data sets were acquired to increase the signal-to-noise ratio and, subsequently, post-processed to calculate the averaged power spectrum of the system response at location S1. The obtained spectrum shows the natural frequencies identified through the experimental modal analysis to be in a reasonably good agreement with those obtained numerically, as illustrated in Table 2.

**Table 2.** Comparison between numerically predicted and experimentally determined flexural resonance frequencies of the composite beam.

| FEM | Experimental |
|---|---|
| 9.3 Hz | 10 Hz |
| 58 Hz | 52 Hz |
| 162 Hz | 145 Hz |
| 316 Hz | 316 Hz |
| 521 Hz | 510 Hz |
| 776 Hz | 760 Hz |

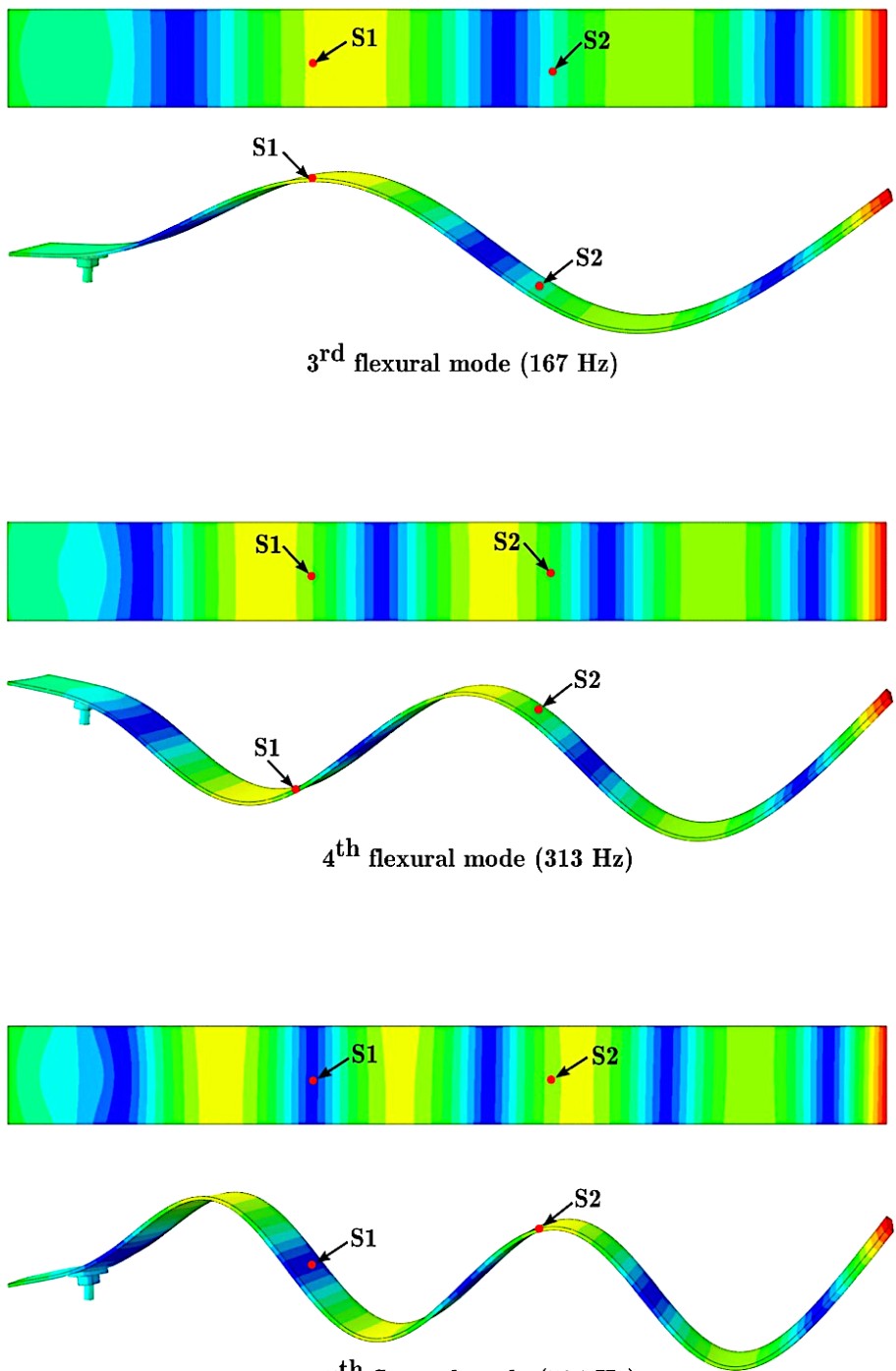

**Figure 4.** Modal shapes corresponding to the 3rd, 4th, and 5th flexural modes predicted by the FE simulations.

The experimental data also confirm that the position S1 corresponds to a node region for the 510 Hz modal shape, as indicated by the antiresonance exhibited by the spectrum at that frequency (Figure 5). To further validate this result, the response of the system was also acquired with the accelerometer placed at a position S2 that lies in the proximity of antinode regions for all the three selected modes shapes (see Figure 4). The spectrum of the system response measured at point S2 presents a clear resonance peak at the 510 Hz frequency (Figure 6), thus providing additional evidence that the point S1 coincides with a node of the 510 Hz mode shape. The three flexural modes associated to the resonance frequencies of 145 Hz, 316 Hz, and 510 Hz obtained by the experimental modal analysis were therefore

selected to investigate the correlation between mode shape and sensor location with respect to the efficiency of the VAM technique.

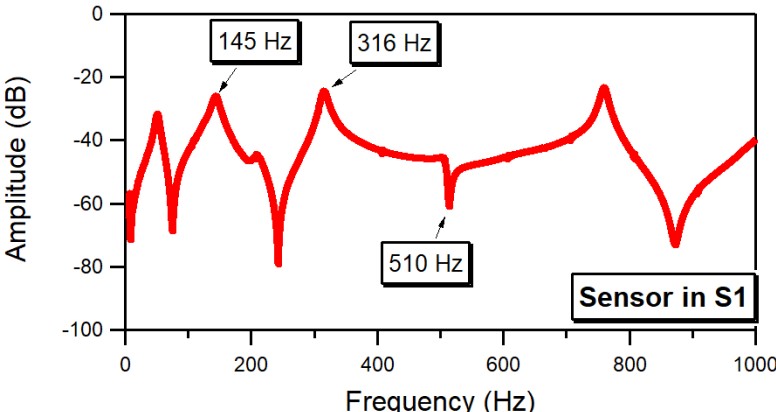

**Figure 5.** Spectral response of the beam in the low frequency range acquired at position S1.

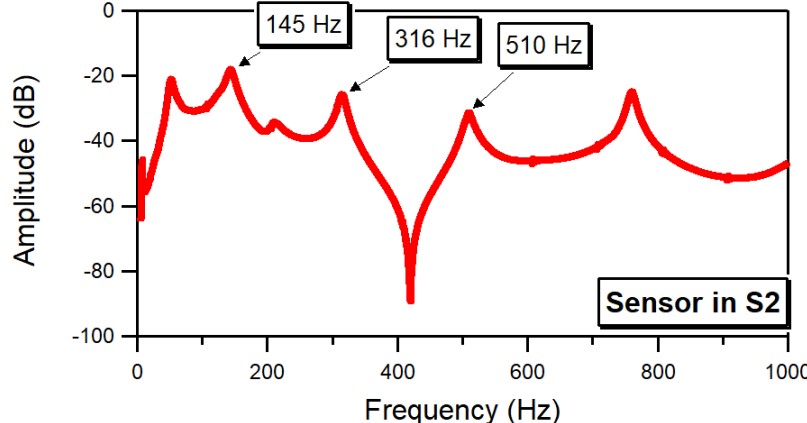

**Figure 6.** Spectral response of the beam at the low frequency range acquired at position S2.

A similar experimental procedure was repeated to characterize the response of the composite beam when subjected to a high frequency excitation. In this case, a linear sweep sine signal varying between 10 and 30 kHz in 6 s was used to drive the low-profile piezoceramic transducer bonded at location A1. The vibration of the system was acquired at the two locations S1 and S2 and then processed to obtain the spectral response of the beam at the two positions. The spectra, shown in Figure 7, show a reasonably wide region of high amplitude values in the proximity of the 21,070 Hz frequency for both the responses measured at the S1 and S2 positions. Therefore, a fixed frequency of 21,070 Hz was selected for the probe excitation of the subsequent VAM tests.

The position S2, which is distant from the nodes for each of the three examined mode shapes (Figure 4), was chosen as the location for the impact, so as to ensure that sufficient strain energy is available at the damage location to activate the nonlinearities generated by the fracture mechanisms.

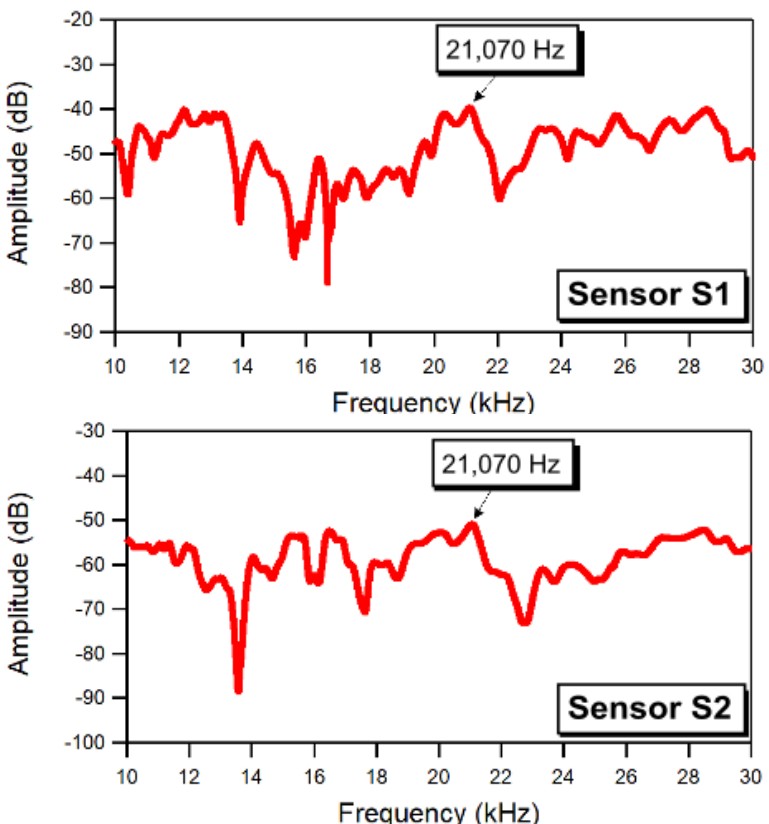

**Figure 7.** Spectral responses of the beam in the high frequency range acquired at positions S1 (**top**) and S2 (**bottom**).

## 5. VAM Tests: Results and Discussion

The VAM tests were conducted using a swept sine signal for the LF pump excitation and a fixed-frequency sine signal for the HF probe excitation. The pump signal consisted of sine sweeps that covered the frequencies of the three flexural modes identified in the experimental modal analyses (145, 316, and 510 Hz). Three series of sweep excitation, in the ranges 105–240 Hz, 255–390 Hz, and 430–565 Hz, and with a duration of 10 s, were applied sequentially to the inspected sample through the electromagnetic shaker. Simultaneously to the swept LF pump excitation, a sinusoidal excitation with a fixed frequency of 21,070 Hz was applied to the composite beam through the PZT stack actuator bonded at point A2 (Figure 2). The amplitudes of the signals supplied to the shaker and to the PZT actuator were respectively 2 and 2.5 Vpp.

The response of the beam was sensed by the accelerometer positioned at location S1 (Figures 2 and 4) and recorded using the PC-controlled oscilloscope at a sampling rate of 400 kSamples/s. The acquired data, averaged over ten data sets, were post-processed by a short-time Fourier transform (STFT) to obtain the time-frequency distribution of the beam response. To this purpose, 110 sub-windows with 50% overlapping were used.

Figures 8–10 show the STFT spectrograms of the response signals acquired in both undamaged and damaged conditions for the three examined frequency spans. The maps plot the amplitude of the signal, represented by a grey intensity level, versus the frequency of the acquired signal (reported on the horizontal axis) and as a function of the instantaneous pump excitation frequency (vertical axis) that corresponds to the specific time of the signal sweep. The maps are zoomed in on a range of frequencies around the 21,070 Hz probe excitation to help visualize the sideband traces. Examples of average spectra extracted at a specific pump frequency (316 Hz) from the spectrograms obtained for the 255–390 Hz swept pump excitation are shown in the bottom row of Figure 9.

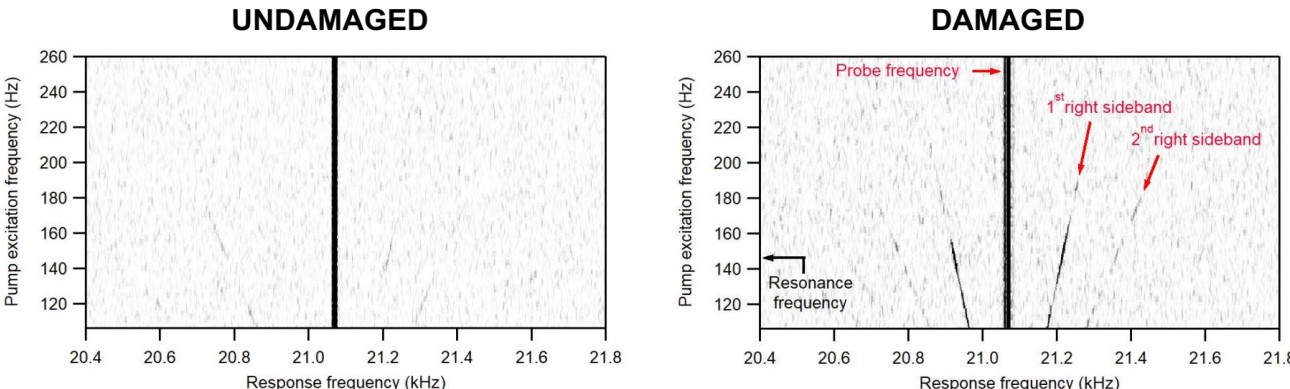

**Figure 8.** Spectrograms of the response of the beam to a swept pump excitation (frequency range 105–240 Hz) in the undamaged (**left**) and damaged (**right**) conditions.

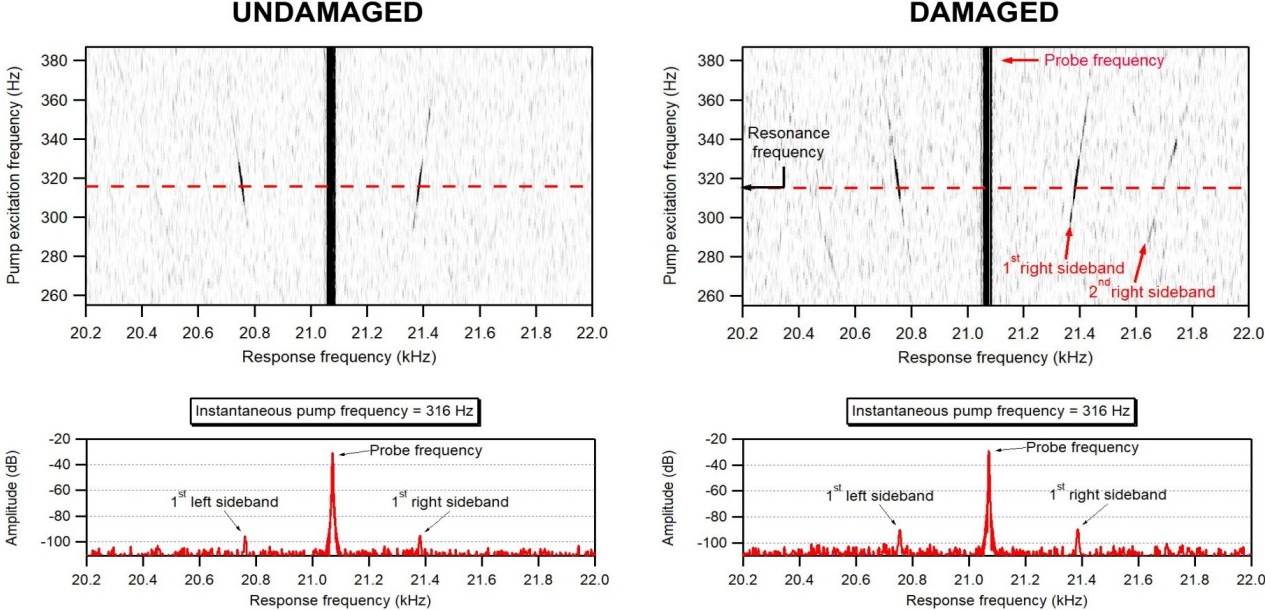

**Figure 9.** Spectrograms of the response of the beam to a swept pump excitation (frequency range 255–390 Hz) in the undamaged (**left**) and damaged (**right**) conditions. The graphs of the bottom row plot the response spectra extracted from the spectrograms at the instantaneous pump frequency of 316 Hz (shown by the dotted red line in the spectrograms).

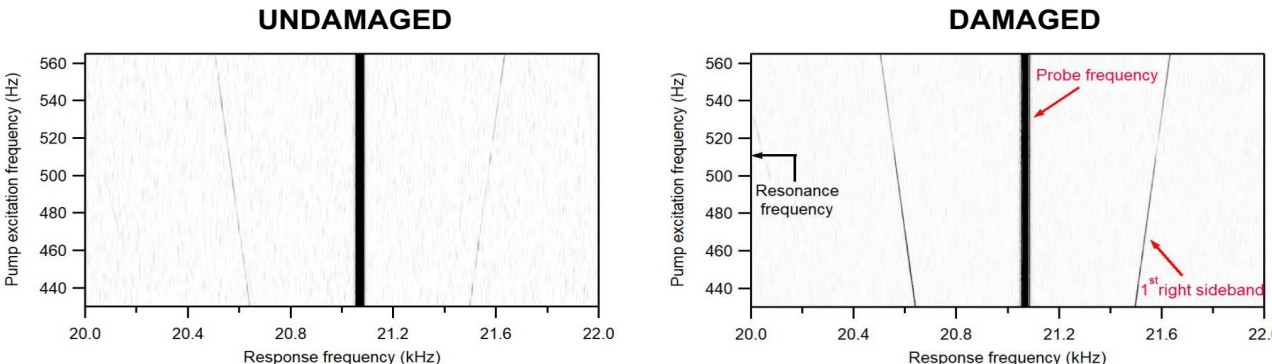

**Figure 10.** Spectrograms of the response of the beam to a swept pump excitation (frequency range 430–565 Hz) in the undamaged (**left**) and damaged (**right**) conditions.

The STFT spectrograms of Figures 8–10 show the presence of modulated spectral sidebands, which are visible as thin straight lines whose distance from the probe frequency (represented by the thick vertical black line) varies linearly with the instantaneous frequency content of the pump excitation. It is worth noting that a second pair of sidebands may be observed in the spectrograms of the beam response acquired for the 105–240 Hz (Figure 8) and 255–390 Hz (Figure 9) pump excitation sweeps.

It is immediately evident that pairs of sidebands are present both in the response of the intact sample and in that of the damaged sample. The presence of sidebands in the response of undamaged materials or samples has been reported in other investigations [13,21,22,33], and may be associated with inherent nonlinearities in the material and in the equipment, or, for this particular setup, with contact effects at the threaded connection between the stud bonded to the beam and the shaker. We can also observe that the amplitude of the sideband depends on the pump excitation frequency, as indicated by the change in grey intensity along the sideband traces. While for the first two frequency spans (105–240 Hz and 255–390 Hz) the sideband amplitude reaches a peak in the proximity of the resonance frequencies (respectively 145 Hz for the first range and 316 Hz for the second range), the sideband response has a minimum at the resonance frequency of 510 Hz in the 430–565 Hz pump frequency range. This is a direct consequence of the position specifically chosen for the sensor, which is placed on a node of the mode shape associated with the 510 Hz but is close to antinode regions of the 145 and 316 Hz modes.

A more careful analysis of the STFT data also shows that the amplitude of the sidebands on the damaged condition is generally higher than on the undamaged condition, as suggested by the darker and sharper sideband traces of the spectrograms of the impacted beam. This is easily noticed when comparing the spectra extracted at specific frequencies of the pump excitation, as visible for example in Figure 9 for the instantaneous pump frequency of 316 Hz. Therefore, the introduction of damage in the composite beam results in an increased amplitude of the sidebands appearing around the probe frequency, even though the extent of this increase appears to be greatly dependent on the frequency of the pump excitation.

In order to provide more quantitative indications on the sensitivity of the technique to the presence of internal damage for different pump frequencies, we further examined the STFT data to compare the changes in the amplitude of the modulation sidebands introduced by the damage as a function of the pump frequency. To this purpose, the three pump frequency ranges were divided into eight sub-ranges with a frequency width of about 17 Hz, and the average magnitudes of the modulation sidebands and of the signal at the probe excitation frequency were calculated for each of these sub-ranges. The extracted values were then used to calculate the modulation index $MI_{sr}$ for the specific frequency sub-range as

$$MI_{sr} = \frac{\sum_i (A_i^L + A_i^R)}{A_{HF}} \tag{5}$$

where $A_i^L$, $A_i^R$, and $A_{HF}$, are, respectively, the amplitudes of the left and right i-th sidebands and the amplitude at the HF probe frequency. Any increase in this index would signal the emergence of additional nonlinear effects and could be used to monitor the onset of new damage in the system. The difference $DI_{sr}$ between the modulation indices $MI_{sr}$ evaluated on the damaged and on the intact beam was then chosen as an indicator of the presence of impact damage:

$$DI_{sr} = MI_{sr-damaged} - MI_{sr-undamaged} \tag{6}$$

The graphs of Figure 11 show the values of the parameter $DI_{sr}$ evaluated at the different sub-ranges of pump excitation frequencies. As expected, since the amplitudes of the sidebands of the damaged beam are larger than, or similar in value to, those of the undamaged beam, the values of $DI_{sr}$ are generally positive over all frequency sub-ranges. The values of $DI_{sr}$ greatly depend on the frequency of the pump excitation. The parameter $DI_{sr}$ typically exhibits relatively low values when the instantaneous frequency

of the pump excitation is distant from the resonance conditions of the beam. The value of $DI_{sr}$ shows, however, a sharp increase around the 145 Hz and 316 Hz resonance frequencies while, conversely, the value of $DI_{sr}$ decreases and reaches a practically null value when the frequency of the pump excitation approaches the 510 Hz resonance frequency. These apparently divergent trends are the result of the position chosen for the measurement sensor, which, as described in the previous section, was purposely placed on a node region for the 510 Hz mode and close to antinode regions for the 145 and 316 Hz modes.

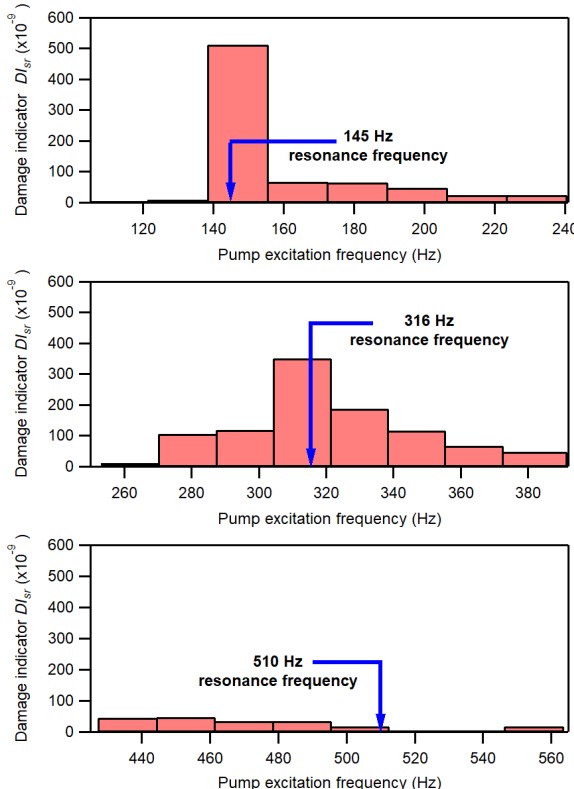

**Figure 11.** Values of the damage parameter $DI_{sr}$ across different sub-ranges of pump excitation frequency.

These results provide clear experimental evidence of the strict link between the sensitivity to damage of the VAM technique and the location of the sensor with respect to the deformation shape activated by the pump vibration. They also indicate that an accurate modal analysis of the system to be monitored is a necessary preliminary step for identifying the best excitation frequencies and the associated optimal sensor placement that would allow to fully exploit the potential of conventional fixed-frequency VAM approaches. It should be also noted that severe damage may significantly change natural frequencies and associated modal shapes of the system, thus affecting the sensitivity of the technique at the selected vibration frequencies.

Alternative and more robust strategies are required to avoid the need of detailed and time-consuming experimental or numerical modal analyses and to improve the reliability of the technique. Monitoring procedures involving the use of multiple-mode or broadband excitations [25–27,31,34,38] have been explored in recent years, even though they have been mainly tested with specific reference to the high frequency probe excitation. The outcomes of this study suggest, however, that the use of a frequency-swept pump excitation may be an effective way to increase the reliability of the indications provided by VAM measurements. As an example, the sum of the $DI_{sr}$ values over each of the three given frequency spans could be adopted as a global damage indicator $DI$, thus providing an assessment of the increase in the nonlinear effects over the examined range of vibrating frequencies. Figure 12 presents a comparison between the values of $DI$ calculated for each

of three examined frequency spans (105–240 Hz, 255–390 Hz, and 430–565 Hz) and those of $DI$ evaluated over a 1 Hz range around the resonance frequency included in the examined frequency span: $DI_{res\ (145)}$, $DI_{res\ (316)}$, $DI_{res\ (510)}$.

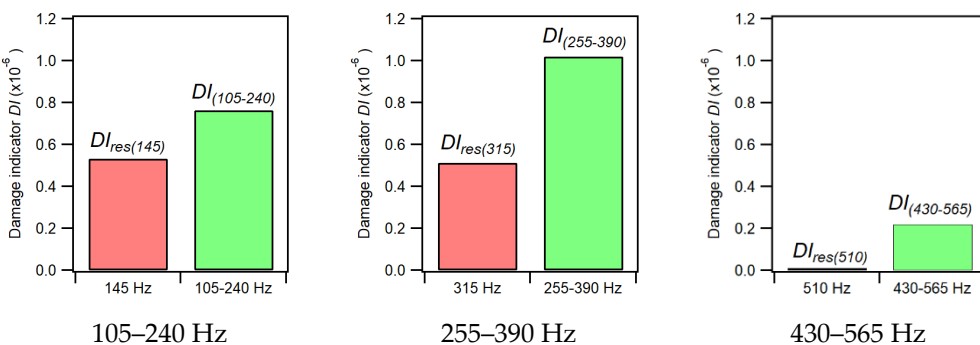

**Figure 12.** Comparison between the damage parameter $DI$ evaluated at a resonance frequency (red bar) and that evaluated as the sum over the eight sub-ranges of the examined pump frequency spans (green bar).

As a final illustration, the cumulative damage parameter $DI_{cum}$ defined as the sum of the $DI$ values evaluated over the three pump frequency spans

$$DI_{cum} = DI_{(105-240)} + DI_{(255-390)} + DI_{(430-565)} \tag{7}$$

is compared in Figure 13 with the sum of the $DI_{res}$ values calculated at the three resonance frequencies

$$DI_{res\_cum} = DI_{res\ (145)} + DI_{res\ (316)} + DI_{res\ (510)} \tag{8}$$

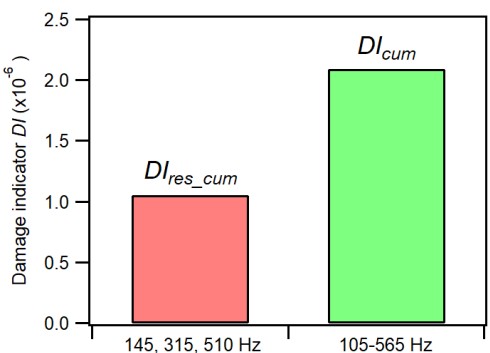

**Figure 13.** Comparison between the damage parameter $DI$ evaluated as the sum over the three resonance frequencies (red bar) and that obtained as the sum over the three frequency spans.

The use of a cumulative damage parameter evaluated over sufficiently wide ranges of pump excitation frequencies is thus shown to increase the robustness and sensitivity of the VAM approach without any need for a detailed previous characterization of modal vibration shapes and natural frequencies of the system.

## 6. Conclusions

The study investigated some aspects related to the choice of the low-frequency pump signal and to the position of the acquisition sensor, which can play a decisive role in the effectiveness of the VAM technique for damage detection in composite materials. The investigation was carried out on a laminated composite beam, initially in pristine (undamaged) condition, and then damaged by a low-velocity impact.

A preliminary finite element analysis was performed to examine the modal shapes related to the first natural frequencies. This allowed for the identification of three flexural

frequencies of the beam (145 Hz, 316 Hz, and 510 Hz), suitable for placing the sensor in nodal or antinodal positions. A frequency-swept pump signal over three frequency ranges covering the three selected resonance frequencies was then applied together with a single harmonic probe wave.

The main indications that arose from the present investigation are summarized below:

(1) The choice of the pump frequency has a critical influence on the sensitivity to damage of the VAM technique, as already shown by previous studies.

(2) Clear experimental evidence shows that the presence of damage may remain undetected when the sensor is placed close to a node region, thus highlighting the strict link between the sensitivity to damage of the VAM technique and the location of the sensor with respect to the deformation shape activated by the pump vibration.

(3) As a direct consequence of this link, a numerical modal analysis of the system is a required preliminary step to identify an appropriate sensor placement when a fixed-frequency pump excitation is applied.

(4) Applying a frequency swept pump vibration together with a fixed-frequency probe excitation was found to be a viable way to improve the sensitivity of the VAM technique for detection of impact damage in the examined composite laminate. In fact, this approach (i) avoids the need for a preliminary numerical modal analysis; (ii) eliminates the risk of selecting ineffective combinations of pump frequency and sensor position; and (iii) allows to cumulate the effects of the inelastic features activated at different resonant frequencies on the response of the system. The use of a cumulative damage parameter evaluated over sufficiently wide ranges of pump excitation frequencies can therefore be used to improve the robustness and sensitivity of the VAM approach for identification of impact damage.

The study shows that the adoption of appropriate multi-modal excitation strategies makes the VAM technique an effective and robust method to reveal the presence of small, localized damage, which is the typical kind of damage occurring in composite structures subjected to out-of-plane impact loads. Nonetheless, further experimental analyses are required to assess its performance across different boundary, structural and environmental conditions. Moreover, the sensitivity of the technique should be also checked in the presence of distributed damage, such as diffused matrix cracking or free edge delaminations, which characterize the response of composite laminates under in-plane static or fatigue loadings.

**Author Contributions:** Conceptualization, F.A., M.C.P. and G.L.; methodology and validation, F.A., M.C.P. and G.L.; investigation, F.A., M.C.P. and G.L.; writing—original draft preparation, F.A., M.C.P. and G.L.; writing—review and editing, F.A., M.C.P. and G.L. All authors have read and agreed to the published version of the manuscript.

**Funding:** This research received no external funding.

**Data Availability Statement:** The data presented in the paper are available upon request.

**Conflicts of Interest:** The authors declare no conflict of interest.

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
