# Peer review of "Influence of Sensor Position and Low-Frequency Modal Shape on the Sensitivity of Vibro-Acoustic Modulation for Impact Damage Detection in Composite Materials"

_jcs, doi:10.3390/jcs6070190_

Round 1

Reviewer 1 Report

The article covers the topic of structural health monitoring for impact damage detection in composite materials using vitro-acoustic modulation. Impact damage is an important challenge in the field of composite materials. Nondestructive methods for detection of such damage are widely researched and urgently needed.

The paper is well written, the introduction covers the topic of vitro-acoustic modulation comprehensively.

However, the manuscript has a few flaws which should be improved:

The approach using a variable pump frequency to avoid the need for FEM analysis for correct positioning of sensors (not in a node of the pump mode) is somehow trivial, as it ensures that the resonances of the structure with different node position are exactly captured.

The difference found for the undamaged and damaged structure is very low to conclude the usefulness of the method at all. Especially since the instrumentation of the sample with sensors (most important S2 directly at the position of the defect) induces an asymmetry itself.

Repeating the experiment with severals structures (composite material is inhomogeneous itself) with different sensor configurations and performing statistics would enhance the significance of the work tremendously.

The damage pattern of the BVID looks strange to me: for an impact damage with 1.9 J no fibre breakage would be expected for specimens with this thickness. The impact pattern looks more like inter fiber fracture in two adjacent layers which wouldn't affect the stiffness of the sample significantly. Why was no high resolution CT performed?

Reviewer 2 Report

This manuscript investigates the detection of impact damages in composite plates by using the vibro-acoustic modulation technique. It is found that by monitoring the amplitudes of the side-band harmonics in a predetermined plate positions it is possible to detect the damage. Experimental measurements are carried out to support the idea. The paper has been written clearly and the focus has been nicely explained via thorough background review. In general, I support its publishing, however the applicability of the method in practice might be questionable. Comments:

1. It is shown in the experiment that the side-band components already exist in the undamaged beam and the damage only increases their amplitudes without exciting any new harmonics. How can you be sure in practice that this is only due to the damage not the other effects such as stiffness reduction, aging, etc?

2. How do you comment its applicability in practice? It is very difficult to control the modal behavior in structural elements and to get a reliable baseline with suitable side-band harmonics. 

In abstract health monitoring should be structural health monitoring.

For clarity, S1 and S2 should be already explained in section 3.

Round 2

Reviewer 1 Report

In their reply, the authors have taken a stand on all points of criticism. however, they have only implemented a few of these points of criticism in their manuscript. The central criticism of checking the method with only one sample and one delamination damage was not included. furthermore, there are still some inconsistencies in the amended manuscript. In lines 243 ff the authors write that only a small damage without fibre breaks was introduced, in Fig. 3 fibre breaks are still drawn in (but they are certainly not fibre breaks!).

Translated with www.DeepL.com/Translator (free version)

Reviewer 2 Report

The authors have answered the questions and improved the paper. Accept.

Author Response

We thank again the reviewer.

Round 3

Reviewer 1 Report

There are numerous studies and papers on low LVI and BVI in composite materials. Because of the energy range, the overall damage extent and the damage pattern I suppose, there is no fiber breakage in the sample. This assumption is also supported by the pattern of the contrast agent uptake for the contrast enhanced radiography.